# Why Are There So Few FDA-Approved Therapeutics for Wound Healing?

**DOI:** 10.3390/ijms242015109

**Published:** 2023-10-12

**Authors:** Mei Chen, Cheng Chang, Brandon Levian, David T. Woodley, Wei Li

**Affiliations:** Department of Dermatology, USC-Norris Comprehensive Cancer Center, University of Southern California Keck Medical Center, Los Angeles, CA 90033, USA; chenm@usc.edu (M.C.); balevian@usc.edu (B.L.);

**Keywords:** wound healing, therapeutics, builder vs damager

## Abstract

Since the only and the milestone FDA approval of becaplermin gel (Regranex^TM^, 0.01% human recombinant PDGF-BB) as a (diabetic) wound healing therapeutic more than 25 years ago, no new therapeutic (excluding physical therapies, devices, dressings, anti-microbial agents, or other preventive treatments) for any type of wound healing has advanced to clinical applications. During the same period of time, the FDA has approved additional 250 new drugs for various human tumors, which were famously described as “wounds that do not heal”. Two similar pathological conditions have experienced such a dramatic difference in therapeutics. More surprisingly, few in the wound healing community seem to be alarmed by this mysterious deficit. As it is often said, “damaging is far easier than re-building”. In contrast to the primary duty of a cancer drug to damage a single molecule of the signaling network, a wound healing drug must be able to re-build the multi-level damages in the wound. No known single molecule alone is capable of repairing multi-cell-type and multi-pathway damages all at once. We argue that the previous single molecule-based strategy for developing wound healing therapeutics is profoundly flawed in theory. The future success of effective wound healing therapeutics requires a fundamental change in the paradigm.

## 1. Wound Healing Is an Essential Recovery Phase for All Human Disorders, Yet It Receives Minimal Support from the NIH and Society

Injuries are a constant occurrence throughout life, affecting various parts of the body at any given time. The healing of moderate tissue damage is typically successful through primary or secondary intention, particularly in healthy individuals. Severe tissue injuries, like second or third-degree burns covering a significant portion of the body, necessitate medical interventions. Inadequate or delayed healing of the injury can lead to systemic chronic inflammation (SCI), which in turn can result in fibrosis, altered tissue function, and, in severe cases, mortality. Several risk factors contribute to the development of SCI, such as infection, physical inactivity, poor diet, exposure to environmental and industrial toxicants, and even psychological stress. SCI is associated with heart disease, diabetes mellitus, cancer, arthritis, inflammatory bowel diseases (IBD), like Crohn’s disease and ulcerative colitis, autoimmune diseases, and neurodegenerative disorders. SCI-associated diseases collectively represent the leading causes of disability and mortality worldwide [1]. Due to interruptions in the normal wound healing process, conditions, such as liver cirrhosis, pulmonary fibrosis, chronic kidney disease, and cardiovascular disease, which together account for around 45% of deaths in developed nations, arise as a consequence of SCI [2]. For example, IBD, which include Crohn’s disease and ulcerative colitis, impacts up to 1.6 million individuals in the United States, including 80,000 children, and over 6 million people globally. As many as 70,000 new cases of IBD are diagnosed in the United States each year. The primary challenges for patients with IBD typically involve the chronic inflammation of all or specific segments of the digestive tract, along with the presence of non-healing and infected wounds. Chronic skin wounds impact millions of Americans with a disproportionate impact on sick veterans, the elderly, and individuals with a lower socioeconomic status. Overall, there are limited effective treatments or cures for disorders associated with chronic wounds. When considering the expenses related to traumatic, burn, chemical, infectious, radiation, and post-surgical wounds, wound healing clinics collectively impose a financial burden on the healthcare system, amounting to hundreds of billions of dollars annually [3]. Even more notably, clinical data concerning certain chronic wounds, such as neuropathic diabetic foot ulcers (DFUs), have indicated a 5-year mortality rate even higher than that observed in many common cancers [4,5,6]. However, among the 350 million people in the US, few have knowledge about the health, social, and economic burdens of DFU, as follows: (i) the number one cause of diabetes-associated hospitalizations, (ii) the primary reason for 45–70% of lower limb amputations, affecting over 100,000 patients annually in the United States, and (iii) a leading factor contributing to a leg being lost every 30 s globally [7,8,9]. Nevertheless, DFUs only represent fewer than 10% of chronic skin wound cases in the US. At the congressional level, wound healing is not listed as a specific area of research by the NIH (National Institute of Health). Wound healing accounts for approximately 0.1% of the NIH budget and 0.15% of the total medical research funded by the non-NIH federal sectors [10,11,12]. To date, the number of effective wound healing therapeutics of any type is nearly nonexistent.

## 2. Wound Healing Is One of the Most Complex Biological Processes during Adult Life

Wound healing is one of the most complex pathophysiological processes in humans. Wound healing is achieved through the temporal and spatial participation of multiple cell types, tissue components, and circulating factors. A malfunction at any single step could disrupt the entire healing process. Most people are not aware that the full completion of acute and normal skin wound healing in humans is more than a year-long process. Using normal skin wound healing as an example, it involves several overlapping phases, including blood clotting (hemostasis), inflammation (immune response), tissue growth (new ECM deposition, cell migration, and proliferation), and dermal tissue remodeling (ECM replacement and neovascularization) [13,14]. Specifically, the primary goal of the **hemostasis phase** is to stop bleeding. Tissue injury leads to the breakdown of blood vessels, extravasation of blood constituents, and blood coagulation, which in turn provides provisional extracellular matrices for epidermal keratinocytes to attach to and migrate on almost immediately following injury. Coagulation and activated-complement pathways also release vasoactive mediators and chemotactic factors that facilitate the recruitment of immune cells to the site of injury. During the inflammation phase, neutrophils and macrophages infiltrate the wound within 24 h of injury to prevent infection. The infiltrated and activated macrophages release various growth factors to stimulate dermal fibroblasts to synthesize and deposit ECMs to form granulation tissue that provides final pavement for keratinocyte migration to close the wound. The monocyte- and macrophage- derived growth factors are necessary for the initiation and propagation of new tissue formation in wounds, as well as the transition from inflammation to the repair phase, albeit with opposite results in different reports. This phase lasts four to six days and is often associated with edema, erythema, heat, and pain. Re-epithelialization through the lateral migration of keratinocytes at the wound edge is the main goal during the **proliferative phase**, the most important phase of wound healing. The process of re-epithelialization begins shortly after the injury, often within a few hours, and continues until the wound is completely closed, which can take place over a span of days to weeks. When keratinocytes migrate, they utilize newly expressed surface integrins as their “feet” to adhere to the underlying ECMs. However, achieving the maximum migration speed for these cells also requires stimulation from soluble factors present in the surrounding environment, although there is ongoing debate regarding the identities of these factors (see the growth factor section below). As the keratinocytes at the wound’s edge advance, they disrupt the cell–cell connections with the keratinocytes positioned behind them. This disruption eliminates cell–cell contact inhibition, prompting the keratinocytes behind to activate the proliferation program to fill the vacated space. Under physiological conditions, the processes of keratinocyte migration and proliferation, in conjunction with wound contraction, collectively account for 85% and 15% of the final wound closure, respectively. The above three phases typically conclude within a few weeks and are collectively referred to as the “wound closure phase”. This phase is widely acknowledged as the tangible outcome of wound healing research and clinical trials. The US FDA’s definition for complete wound closure is 100% wound surface re-epithelialization without discernible exudate, drainage, and dressing requirements confirmed at two consecutive study/clinical visits two weeks apart. The final **remodeling phase** following wound closure lasts over a year, during which the new tissue slowly gains strength and flexibility. Any additional information provided regarding this phase of wound healing is purely speculative, as there have been limited animal models and methodologies available to monitor the entire duration of the wound remodeling phase. The complexity of wound healing stems from the fact that each of the four phases of wound closure involves its own multiple cell types, specific ECMs and integrin receptors, and different microenvironmental factors. Using the proliferation phase of wound healing as an example, several processes occur concurrently. This includes angiogenesis, the formation of new blood vessels by vascular endothelial cells; collagen deposition, where fibroblasts expand and deposit provisional collagen and fibronectin; granulation tissue formation; epithelialization, with keratinocytes migrating and proliferating to close the wound; and wound contraction, where myofibroblasts reduce the size of the wound by contracting it. Additionally, any surplus myofibroblasts that are no longer needed undergo apoptosis.

Finally, wound healing is repair with little regeneration in humans. Regeneration is known as the natural process of replacing or restoring damaged or missing cells, tissues, or organs to their original and fully functional states in plants and animals. Due to the complexity of mammals, most organs are incapable of regeneration, with the exceptions of such appendages, such as the hair, nails, the top epidermal layer of the skin, the liver, and stomach (following surgery). Therefore, wound healing is not a regenerative process and will not give rise to the original architecture before the injury occurred. Instead, the primary objective of wound healing is to repair and restore tissue homeostasis while minimizing water loss, salt loss, and the risk of life-threatening infections [13]. A recently healed skin wound, even in the case of a healthy adult, cannot fully revert to its original, unwounded state and typically loses many skin appendages, such as hair follicles, sweat glands, and sebaceous glands, which are often replaced by scar tissue. Due to space limits and the specific focus of this article, we refer readers to three excellent reviews, in chronological order, on the mechanisms of wound healing in more detail [13,14,15].

## 3. A Profound Distinction between a “Wound Healing Driver Gene (WDG)” and “Cancer Driver Gene (CDG)”

Numerous genes have been identified as essential for various pathophysiological processes, but only a limited number of them have been found to be individually capable of driving these processes, such as driving wound healing and cancer progression into completion. A cancer driver gene (CDG) is defined as a gene that, when mutated, increases net cell growth within the specific in vivo microenvironment [16,17]. Based on the “Two-Hits” [18] and “Multi-Hits” [19,20] theories of cancer causation, no single oncogene alone possesses the capacity to initiate cell transformation and sustain the entire tumorigenic process to completion. Rather than being driven by a single cause, cancer arises from the accumulation of multiple mutations that are randomly induced in genes responsible for regulating normal cell growth. These mutations can affect either growth control genes or growth-suppressor genes, both of which play crucial roles in regulating cellular growth. In other words, individual oncogenes or tumor suppressor genes are crucial components, but on their own, they lack the capacity to independently produce cancer. A typical example is the ras oncogene (v-ras), which has been detected in more than 30% of solid tumors in humans. The overexpression of v-ras was unable to transform primary cells; instead, it caused cell senescence. The activation of a single oncogene or inactivation of a single tumor suppressor gene by itself is not sufficient to drive cancer progression [21,22,23,24,25]. On the other hand, since each of these CDG is a key component of the much larger signaling network in tumor cells, the loss of function of any of these CDG could partially or completely impair the entire signaling network, resulting in the inhibition of cancer progression. Taking a scaffold structure of a building as an intuitive comparison, missing any individual tube or coupler may cause a catastrophic collapse of the entire scaffold structure. CDG can be likened to these individual tubes or couplers within the scaffold-like signaling network inside tumor cells. This understanding of CDG in cancers has resulted in the approval of over 1000 anti-cancer drugs by the FDA (as of the end of 2021). These drugs work by inhibiting, neutralizing, or eliminating the functions of individual CDG.

The term “wound healing driver gene” (WDG) was first proposed by Tang and colleagues [12]. However, the definition of a WDG is fundamentally different from a CDG. Instead of the single requirement necessary for a CDG during tumorigenesis, a WDG must satisfy the dual criteria of being both necessary and sufficient to orchestrate the entire signaling network and bring about the completion of the multi-phase wound healing process. Hence, it is improbable for a WDG to be a single gene product, given the intricate temporal and spatial involvement of multiple cell types throughout the wound healing process. Specifically, no single tissue component, growth factor (cell type restricted), ECM (integrin-specific), or cell type (event-specific) is sufficient on its own to complete the wound healing process from A to Z [14]. In theory, a collective combination of all the individual components, akin to the “tubes” or “couplers” within the entire scaffold structure, working together may show the potential to fulfill the sufficiency criterion for a WDG. Therefore, a WDG refers to a combination of all essential factors, including those that stimulate dermal fibroblasts to deposit ECMs for pavement, those that drive keratinocyte migration and proliferation to close the wound, and those that re-build the blood circulation system to support oxygen and nutrients. In our opinion, this fundamental difference is the direct cause for the current 1000-to-1 ratio between FDA-approved cancer to wound healing drugs. Over the past decades, the wound healing community has primarily employed a strategy reminiscent of the development of anti-cancer drugs by focusing on single-factor-based therapeutic approaches. Unlike drug targets for cancer, however, many pharmaceutical companies perceive the wound repair process as excessively intricate and investing in the development of wound healing drugs as carrying a much higher level of risk. For instance, while an acute skin wound heals in 4 to 6 weeks, chronic wounds fail to heal within the timeframe and remain open for months or even years. There is no known single-molecule “builder” capable of repairing these wounds [12].

## 4. A Growth Factor Represents the Classical Example of a Non-Builder for Wound Healing

Since the discovery of growth factors in the 1970s, the prevailing belief has been that growth factors are the driving force for skin wound healing [26,27]. More than 30 growth factors have been subjected to extensive clinical trials alone or in combination, including epidermal growth factor (EGF), basic fibroblast growth factor (bFGF), acidic FGF (aFGF), granulocyte and macrophage colony-stimulating factor (GM-CSF), and platelet-derived growth factor-BB (PDGF-BB). Although some of the double-blinded trials reported “promising” clinical efficacies [28,29,30,31,32,33,34,35,36,37,38,39,40,41,42], only human recombinant PDGF-BB received approval from the US FDA in 1997 (Regranex^TM^/becaplermin gel, Ortho-McNeil Pharmaceutical, Raritan, NJ, USA) for restricted clinical applications. Becaplermin gel, a topical gel labeled for the adjunctive treatment of stage III and IV DFUs of the lower extremities, operates by enhancing granulation tissue formation. Non-surgical treatment options mainly include simple daily wound care, using becaplermin gel or skin substitutes, to promote wound closure, and antibiotics to treat deep infection, drainage, and cellulitis. Surgical treatment choices primarily involve debridement, which is performed to promote blood flow. Becaplermin gel is not recommended for the topical treatment of any other types of chronic and acute skin wounds due to a lack of clinical data, as well as cancer-causing concerns. After approval in 1997, subsequent larger-scale, multi-center, randomized parallel trials showed that the becaplermin gel improved diabetic wound closure by 14 compared to the placebo (50% improvement versus 36% with placebo control). If this data were available prior to its initial FDA approval, this level of improvement might not have been considered cost-effective for clinical practice. The formulated concentration of PDGF-BB in becaplermin gel is at least 1000-times greater than the physiological PDGF-BB level in human circulation (0–15 ng/mL), which once caused a “black box” label over an increased risk of cancer mortality in patients who require treatments of three (15 g) tubes or more [43,44,45,46,47]. Our laboratory has investigated and identified two main reasons as to why PDGF-BB has not lived up to its expected level of effectiveness. First, only dermal fibroblasts express PDGF-BB receptor. However, keratinocytes and microvascular endothelial cells, responsible for wound closure and the formation of new blood vessels, respectively, lack the PDGF-BB receptor [48]. Second, the increased concentration of TGFβ3 in the wound environment completely nullifies the effectiveness of PDGF-BB [49]. In addition, PDGF-BB and other growth factors cannot be natural wound healing-promoting factors since broken blood vessels and blood clotting immediately following wounding prevent the continued supply of growth factors into the wound bed. In addition to growth factors, many other biomaterials, such as angiogenic factors and ECM, have been studied or entered clinical trials for wound healing, for which we recommend two recent excellent reviews [50,51].

## 5. The Fundamental Reason behind So Many Cancer Drugs and So Few Wound Healing Therapeutics: It Is Always Easier to “Damage” Than to “Re-Build”

It is widely recognized that wound healing and tumor progression share common events and microenvironments, albeit opposite desired outcomes. However, the requirements for making a cancer drug and for making a wound healing drug are disproportionally unfair to the latter. The fundamental difference between a cancer drug, for which the FDA-approved number has reached over 1000 (by end of 2021), and a wound healing drug, for which the FDA-approved number remained unchanged since 1997, is the difference between a “damager” versus a “builder”. It is far easier to damage than to re-build. The primary requirement for an anti-cancer drug is to function as a damager or disruptor, affecting a critical element of cellular events, like survival, growth, or invasion, while leaving all other components within the same signaling network unharmed. As illustrated in Figure 1, drugs that obstruct or induce damage in any of the numerous events that support tumor growth, including angiogenesis, tumor metabolism, immune checkpoints, DNA stability via oncogenes, or tumor suppressor genes, have the potential to impede or halt the progression of the tumor. The 2022 10 top-selling cancer drugs that collectively make more than $70 billion in sales are all single CDG-based blockers inside the signaling network of a tumor. These drugs include the following: (#1) Revlimid (lenalidomide) by Celgene has been on the market for a decade (2022 sales: $13.44 billion) and has been used to treat multiple myeloma, myelodysplastic syndromes, and mantle cell lymphoma. Revlimid is a small molecule derivative of thalidomide that targets E3 ubiquitin ligase, causing the ubiquitination and degradation of the IKZF1 and IKZF3 transcription factors. (#2) Opdivo (nivolumab) by Bristol-Myers Squibb and Ono Pharmaceutical (2022 sales: $12.62 billion) has been used to treat non-small cell lung cancer, metastatic melanoma, renal cell carcinoma, and classical Hodgkin lymphoma. Nivolumab is a fully human immunoglobulin G4 PD-1 immune checkpoint inhibitor antibody and blocks PD-1 and promotes antitumor immunity. (#3) Imbruvica (ibrutinib) by AbbVie (Pharmacyclics) and Johnson & Johnson (2022 sales: $8.29 billion) has been used to treat chronic lymphocytic leukemia, mantle cell lymphoma, and Waldenström macroglobulinemia. It is a small molecule inhibitor that blocks Bruton’s tyrosine kinase (BTK) in B cells to stop B cells from surviving and multiplying. (#4) Keytruda (pembrolizumab) by Merck & Co. (2022 sales: $6.56 billion) is the first humanized antibody targeting PD-1 and has been used to treat advanced melanoma, non-small cell lung cancer, and head and neck squamous cell cancer. (#5) Ibrance (Palbociclib) by Pfizer (2022 sales: $6.01 billion) is a small molecule inhibitor of the cyclin-dependent kinases CDK4 and CDK6 and has been used to treat metastatic breast cancer. (#6) Tecentriq (atezolizumab) by Roche (2022 sales: $5.53 billion) is a monoclonal antibody that binds PD-L1 on the surface of cancer cells and prevents cancer cells from suppressing the immune system. Atezolizumab has been used to treat urothelial carcinoma and non-small cell lung cancer. (#7) Darzalex (daratumumab) by Johnson & Johnson (2022 sales: $4.91 billion) is a monoclonal antibody medication that binds to CD38, which is overexpressed in multiple myeloma cells. It induces the killing of multiple myeloma and other hematological tumors. (#8) Perjeta (pertuzumab) by Roche (2022 sales: $4.73 billion) is a monoclonal antibody targeting the HER2 receptor tyrosine kinase. It is used in combination with trastuzumab and docetaxel for the treatment of metastatic HER2-positive breast cancer. (#9) Xtandi (enzalutamide) by Astellas Pharma and Pfizer (2022 sales: $4.71 billion) is in a class of medications called androgen receptor inhibitors. It works by blocking the effects of androgen (a male reproductive hormone) to stop the growth and spread of prostate cancer. (#10) Avastin (bevacizumab) by Roche (2022 sales: $4.68 billion) is a monoclonal antibody-based angiogenesis inhibitor. It works by blocking the function of the vascular endothelial growth factor A (VEGF-A) and the growth of blood vessels that are needed to support tumor growth. It is recommended to treat colorectal cancer, non-small cell lung cancer, ovarian cancer, cervical cancer, renal cell carcinoma, and glioblastoma. All of these drugs target a single signaling molecule either inside the tumor cell or its microenvironment.

In contrast to cancer drugs as a damager, a wound healing drug must act as a “builder”. It should possess the capability to repair the various layers of damage within the wound bed while simultaneously initiating multiple crucial signaling events, such as new ECM deposition by dermal fibroblasts as pavement, keratinocyte migration and proliferation on the pavement to close the wound, and endothelial cell neovascularization to provide oxygen and nutrients. If wound repair were compared to building a house, the single builder would have to be a qualified general contractor, a concrete expert, a carpenter, a roofer, a plumber, an electrician, a drywall expert, and a painter all at once. As depicted in Figure 2, to the best of our knowledge, there is no single known molecule that fulfills all these diverse requirements of a builder to promote wound closure. Single molecule-based therapeutics might address a specific type of damage within the wound bed. However, relying solely on repairing individual damaged events, as the previous decades-long approach to wound healing drug development, is unlikely to generate clinical significance. Thus, in theory, the remedy for wound healing will have to be a combination of multiple factors or cell types. Furthermore, the solution must include the mechanism that enables these components to work together in a coordinated manner, both spatially and temporally, to effectively repair the multi-tiered damages within the wound. Currently, neither the complete list nor the actions of these hypothetical factors for wound closure is available. Nonetheless, even if all the relevant components of the “builder” and their mechanisms of action were identified, few pharmaceutical companies would take it as a viable drug development project. Each new molecule drug candidate would have to go through its own FDA-demanded toxicity, pharmacodynamics, pharmacokinetics, and efficacy trials in humans. These new trials often take years, cost billions of dollars, and yet culminate in a notably low success rate, which even for cancer drugs, is less than 5% [52]. Attempting to develop a comprehensive and multi-component remedy all at once is impractical. What new approaches can be considered to address these challenges? First, we predict that natural and multi-factor tissue biologics, which have lower hurdles to obtain FDA approvals than devices, are likely to be qualified as the “builder” of wound healing. These tissue biologics are easy to produce, safe, and effective. Among the emerging biologic therapies, for instance, the cryopreserved umbilical cord TTAX01 allograft from Amniox Medical is expected to garner the highest patient share owing to promising efficacy signals shown in phase II trial and ongoing two-phase III multicenter randomized controlled clinical trials (https://clinicaltrials.gov/ct2/show/NCT04176120, accessed on 13 September 2023). Second, overcoming the technical challenges linked to stem cell therapy, including issues, like extended cell survival, improved differentiation, and enhanced communication within the wound, could make stem cell therapy an even more attractive option. Due to the specific focus of this article, we refer readers to several excellent research articles and reviews on wound healing biologics in detail [53,54,55,56].

## 6. What Is the Future for Wound Healing Therapeutics?

Currently the future of wound healing therapeutics is far from clear. Our “damager-builder” theory explains the 1000-to-1 ratio of cancer therpeutics to wound healing therapeutics over the past several decades. While pharmaceutical companies should avoid the development of single molecule-based wound healing therapeutics, stem cells, multi-factor biologics, and even Oriental medicine-based therapies will arrive at wound healing clinics.

## Figures and Tables

**Figure 1 ijms-24-15109-f001:**
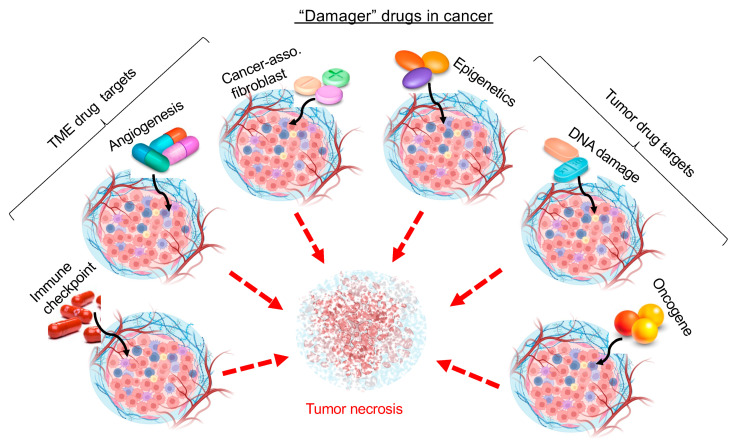
**Easier to make a “damager” drug in cancer.** If any one of the five basic parts in a computer, namely a motherboard, a central processing unit, a graphics processing unit, a random-access memory, and a hard disk or solid-state drive, is damaged, the whole computer collapses. If any of the major organs in the human body is damaged, the life of the human ends. Likewise, if the any one of the six tumor-supporting units, as shown, is targeted and damaged by a drug, the tumor progression stops. Therefore, cancer drug makers have taken a full advantage of the philosophy from ancient Chinese Tai Chi, “四两拨千斤”, meaning “to make great accommodations with (relatively) little efforts”. Any single factor, such as an organic chemical, an antibody, a nucleotide, or a modified gene product with a delivering mechanism, could act as a damager to a tumor.

**Figure 2 ijms-24-15109-f002:**
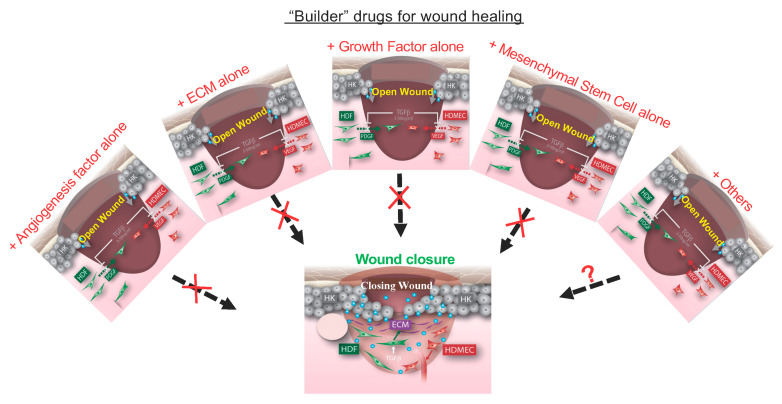
**Impossible to make a “builder” drug for wound healing from a single molecule.** When tissue is wounded, multiple tissue components have been damaged. To repair and re-build the damaged tissue, a healthy body initiates a healing program inside the wound involving multiple cell types, new ECMs, and cell type-specific soluble factors. In contrast, a similar healing program cannot be launched in chronic wounds in humans with various compromised health conditions, and no single factor has the capacity of replacing the healing program. Any single factor can only directly affect one of the many coordinated events, as shown, during wound healing. For instance, the addition of an angiogenic factor activates angiogenesis, but it has little direct effect on the deposition of ECMs by dermal fibroblasts nor keratinocyte migration to re-epithelialize the wound. Similarly, PDGF-BB selectively activates dermal fibroblasts, but nether keratinocytes nor endothelial cells express PDGF receptors. “X” means “no” and “?” means “may be”.

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
