# Peer review of "Why Are There So Few FDA-Approved Therapeutics for Wound Healing?"

_ijms, 2023, doi:10.3390/ijms242015109_

Round 1
Reviewer 1 Report
This manuscript presents a very good revision of the literature in an attempt to explain the scarcity of wound healing treatments that get FDA approval compared to cancer treatments. I really enjoyed reading the paper and I was hoping to see more on current holistic /multifactorial treatments specifically biologics and stem cells. Many of these treatments are currently in clinical trials and the authors should add a paragraph or two detailing these studies.
Author Response
Dear Reviwers:
We thank you all for your overwhelming enthusiasm toward this review article. We have now completed our revision by 1) implementations of the reviewers’ comments and 2) thoroughly editing the English (all in red). For the latter, we have carried out an extensive editing of the English by medical expert with native English speaking at USC! The following is our point-to-point responses to your comments:
Reviewer 1:
Q “I was hoping to see more on current holistic /multifactorial treatments specifically biologics and stem cells. Many of these treatments are currently in clinical trials and the authors should add a paragraph or two detailing these studies. “
Reply: Accepted! Since none of these biologics has advanced to FDA approvals and owing to the space limit, we have instead added two excellent review articles on this issue by others.

Reviewer 2 Report
First of all I would Like to thank the authors for their efforts, but I have a few comments
1- The title should be more convenient and specific
2- The literature review about various techniques and produts approved or not approved by FDA still need update
3- The authors should consider some modern techniques expressing wound dressing materials
4- The authors should More clearly express the history of wound dressings
4- in my opinion some references should be added to the review related to various techniques such as for example
http://dx.doi.org/10.3390/pharmaceutics15051518
http://dx.doi.org/10.3390/polym14030454
Reviewer 3 Report
The manuscript is well written with appriopriate background information and adequate references.
I have only some minor comments.
1. Some abbreviations like DWG are explained many times in the manuscript.
2. There are some highlighted words in the manuscript like significantly etc. Is it in purpose or by chance?jökus
Reviewer 4 Report
The authors should consider the followings:
The authors may broaden their search and compare the results as to EMA, or other appropriate sized agency.
The authors may need to provide evidences to support the use of cancer therapeutics, as the justified comparator for that of wound healing (i.e. in terms of comparable, prevalences, mortality, scores of quality of life, cost in health economics etc).
"no second therapeutic (excluding physical therapies, devices, dressings, anti-microbial agents or other preventive treatments." The authors should explain why they exclude the second therapeutic in the review, in their introduction part.
In general, this review article required more evidences and figures to support the idea of the authors.
Informative charts and tables should be used to enlist the endorsed wound healing products, verus the attempted products, across the years, to demonstrate the comparison.
The authors shall include the relevant and recent literatures of the topic.
The authors can use another table, to further breakdown the types of wound healing agents, by mechanism of action and/or by types.
As for search results from clinical trial databases, the authors should use Informative tables to list the wound healing trials, across the years. The authors may describe the search strategy and clinical trial databases used. Alternatively, the search for cancer therapeutics should be compared, by the relevant databases.
The authors may improve figure resolutions of Figure 2. The in-figure font should be placed horizontal, for easy reading.
Extensive editing of English language required
Reviewer 5 Report
This review highlights the stagnation in wound healing therapeutics compared to cancer treatment progress. It explains the complexity of wound healing involving multiple factors and cell types, suggesting the need for diverse approaches and increased research funding to advance the field. While the concept of this review article holds significant value within the scientific literature, I believe that it would benefit from a more comprehensive discussion to enhance the clarity and readability of the paper.
I have following concerns that need to be addressed:
1. This review lacks references to support the research and statistical data presented at several places. The authors should include appropriate references in these sections to support their claims.
2. Authors should incorporate the most up-to-date research available to date, ensuring they reference the latest articles addressing the current status.
3. The concept of using natural and multi-factor tissue biologics as potential wound healing therapeutics is mentioned. Authors should provide examples of such biologics and explain how they may work to promote wound healing compared to single molecule-based approaches?
4. What are the potential implications of the differences between "damaging" cancer drugs and "building" wound healing drugs on the development process and clinical outcomes? How do these differences influence the strategies and challenges encountered by pharmaceutical companies in developing wound healing therapeutics?
5. Authors should include a dedicated section that discusses current advancements and explores alternative approaches or strategies for wound healing. Additionally, authors may consider discussing strategies or initiatives that could help bridge the gap in research funding and accelerate progress in this field.
Round 2
Reviewer 2 Report
In general the authors have made most of the changes that I asked for and they probably give reasonable answers to my questions.
Reviewer 5 Report
I believe that the authors have properly addressed the main concerns that I had and hence I now do support publication.